# GPS Phase Integer Ambiguity Resolution Based on Eliminating Coordinate Parameters and Ant Colony Algorithm

**DOI:** 10.3390/s25020321

**Published:** 2025-01-08

**Authors:** Ning Liu, Shuangcheng Zhang, Xiaoli Wu, Yu Shen

**Affiliations:** 1College of Geology Engineering and Geomatics, Chang’an University, Xi’an 710054, China; shuangcheng369@chd.edu.cn (S.Z.); sy026@chd.edu.cn (Y.S.); 2Shaanxi Satellite Application Technology Center of Natural Resources, Xi’an 710119, China; xianyangwuxiaoli@126.com

**Keywords:** integer ambiguity, QR decomposition transformation, Kalman filter, Cholesky decomposition, ant colony algorithm, decorrelation processing

## Abstract

Correctly fixing the integer ambiguity of GNSS is the key to realizing the application of GNSS high-precision positioning. When solving the float solution of ambiguity based on the double-difference model epoch by epoch, the common method for resolving the integer ambiguity needs to solve the coordinate parameter information, due to the influence of limited GNSS phase data observations. This type of method will lead to an increase in the ill-posedness of the double-difference solution equation, so that the fixed success rate of the integer ambiguity is not high. Therefore, a new integer ambiguity resolution method based on eliminating coordinate parameters and ant colony algorithm is proposed in this paper. The method eliminates the coordinate parameters in the observation equation using QR decomposition transformation, and only estimates the ambiguity parameters using the Kalman filter. On the basis that the Kalman filter will obtain the float solution of ambiguity, the decorrelation processing is carried out based on continuous Cholesky decomposition, and the optimal solution of integer ambiguity is searched using the ant colony algorithm. Two sets of static and dynamic GPS experimental data are used to verify the method and compared with conventional least squares and LAMBDA methods. The results show that the new method has good decorrelation effect, which can correctly and effectively realize the integer ambiguity resolution.

## 1. Introduction

Precise relative positioning is realized by using double-difference carrier phase data. Due to the data containing integer ambiguity, the distance observation value between satellite and receiver is not complete until the ambiguity is determined. Therefore, integer ambiguity resolution plays an extremely important role in obtaining high-precision GPS positioning results [1,2,3]. In order to verify that the ambiguity can be correctly determined to achieve precise positioning and measurement, and now with the development of On-The-Fly ambiguity resolution technology, there have been many practical ambiguity resolution methods, however, the research on ambiguity resolution is still a hot topic for many scholars. At present, the ambiguity resolution methods are mainly summarized into three categories [4], firstly, the ambiguity is determined in the observation range. Generally, CA code or P code is combined with the phase to determine the ambiguity, due to the limitation of the accuracy of the code observation itself. These methods usually use the linear combination of phase between the frequencies to obtain the integer ambiguity after smoothing [5,6]. Secondly, the ambiguity is determined in the coordinate domain. These methods determine the optimal coordinate estimation in a coordinate search space on the premise that the ambiguity is an integer. The ambiguity function method is a representative of the second type [7,8]. Thirdly, the ambiguity is determined in the ambiguity domain. These methods are the most effective and commonly used method to determine the ambiguity. Its basic idea is to use the least squares or the Kalman filter to estimate the float solution of the ambiguity and construct the candidate space of ambiguity with the float solution, and then, search for the correct integer ambiguity [9]. Based on the third integer ambiguity resolution strategy, some scholars have carried out related research on improving the decorrelation effect and search performance for resolving ambiguity [10,11,12,13,14]. Particularly with the emergence of artificial intelligence optimization algorithms, such as genetic algorithm, particle swarm optimization algorithm is necessary for searching for integer ambiguity [15,16,17].

Ant colony algorithm is a new modern heuristic algorithm proposed by Dorigo for combinatorial optimization problems [18]. The advantages of the algorithm, such as the positive feedback principle, distributed computing, and artificial heuristic information, enable it to quickly obtain a stable solution [19]. Through the analysis of LAMBDA search algorithm, it can be seen that the value of $\chi^{2}$ controls the volume of ambiguity search in the hyper-ellipsoid region. Also, in the whole search process, the volume of hyper-ellipsoid remains unchanged, so its search process cannot be improved, and the search algorithm is more complex. Therefore, in order to enrich the integer ambiguity search method and further improve the success rate of integer ambiguity solutions, the ant colony algorithm is introduced into the integer ambiguity search. Firstly, the coefficient matrix in the GPS double-difference observation equation is decomposed by QR decomposition transformation, so as to eliminate the coordinate parameter information. Then, the Kalman filter is used to estimate the only ambiguity parameter contained in the transformed equation. Based on the float solution of ambiguity obtained by the Kalman filter, continuous Cholesky decomposition is used for decorrelation processing, and the optimal integer solution of ambiguity is searched by combining the ant colony algorithm. The static and dynamic observation data are selected to verify the feasibility and correctness of the new method for integer ambiguity resolution.

## 2. Methods

In this section, the problem of solving ambiguity using conventional GPS double-difference observation equation is firstly analyzed. Next, the positioning equation without coordinate parameters is established by *QR* decomposition transformation, and float solution of the ambiguity is solved using the Kalman filter. Then, the specific steps of ambiguity decorrelation processing based on continuous Cholesky decomposition. The search strategy of solving integer ambiguity using the ant colony algorithm are described in detail.

### 2.1. Calculating Float Solution of Ambiguity Based on Eliminating Coordinate Parameters and Kalman Filter

When two GPS receivers placed on the base station *T*_1_ and the rover station *T*_2,_ respectively, and viewed *n* + 1 satellites in common, each epoch can form *n* phase double difference observation equations, so the linearized double-difference observation equations can be derived as follows [20,21]:(1)∇∆L=AδX+B∇∆N+e

In the above formula, ∇∆L is the double-difference phase observation vector, $\nabla ∆L={\left[\nabla ∆L^{1}\nabla ∆L^{2}\cdots \nabla ∆L^{n}\right]}^{T}$, δX is the coordinate correction vector of rover station *T*_2_, $\;\delta X={\left[\delta X_{2}\delta Y_{2}\delta Z_{2}\right]}^{T}$,∇∆N is the integer ambiguity vector, $\;\nabla ∆N={\left[\nabla ∆N_{1}\nabla ∆N_{2}\cdots \nabla ∆N_{n}\right]}^{T}$ e  is the observation error vector, $e={\left[e^{1}e^{2}\cdots e^{n}\right]}^{T}$, A  and B are the corresponding coefficient matrix, respectively. $A=\frac{-1}{\lambda }\left[\begin{array}{c} \nabla l_{2}^{1}\;\nabla m_{2}^{1}\;\nabla n_{2}^{1}\\ \nabla l_{2}^{2}\;\nabla m_{2}^{2}\;\nabla n_{2}^{2}\\ \cdots \;\cdots \;\cdots \\ \nabla l_{2}^{n}\;\nabla m_{2}^{n}\;\nabla n_{2}^{n} \end{array}\right]$, $B=\left[\begin{array}{c} -1\;0\;\cdots \;0\\ 0-1\;\cdots \;0\\ \cdots \;\cdots \;\cdots \;\cdots \\ 0\;0\;\cdots -1 \end{array}\right]$. Using the least squares estimation for Equation (1), the real number estimate of δX and ∇∆N can be obtained as follows [22]:(2)$$\left[\begin{array}{c} \delta X\\ \nabla ∆N \end{array}\right]={\left[\begin{array}{cc} A^{T}PA & A^{T}PB\\ B^{T}PA & BPB \end{array}\right]}^{-1}\left[\begin{array}{c} A^{T}P\nabla ∆L\\ B^{T}P\nabla ∆L \end{array}\right]$$
where P is the weight matrix. By analyzing Equation (2), it can be seen that in the process of solving the above normal equation by epoch using the least square method, due to the limited GPS phase observation, the ill-condition of the double-difference solution equation will increase. The conventional method is to increase the number of epochs or introduce corresponding pseudorange observation to reduce the ill-posedness and solve it. However, then the problem of high-order matrix multiplication and inversion will be encountered, which increases the computational complexity and cannot meet the requirements of rapid processing of GPS data. In the initial stage of relative positioning, the main purpose is to solve the float solution of ambiguity and its variance matrix. Therefore, *QR* decomposition can be applied to the coefficient matrix *A*, and the decomposed *Q* matrix can be divided into blocks, that is $Q=\left[\begin{array}{cc} \underset{n\times 3}{\underbrace{Q_{1}}} & \underset{n\times (n-3)}{\underbrace{Q_{2}}} \end{array}\right]$, the block submatrix $Q_{2}$ is transposed; the transposed matrix $Q_{2}^{T}$ is used to multiply both sides of Equation (1), because of the relationship [23] $Q_{2}^{T}A_{i}=0$, a new observation equation without the coordinate correction vector can be obtained as follows:(3)$$Q_{2}^{T}\nabla ∆L=Q_{2}^{T}B\nabla ∆N+Q_{2}^{T}e$$

It can be seen from Equation (3) that the dimension of the unknown parameters is reduced after the above transformation, and the float solution of ambiguity can be calculated even if the single epoch phase data are used.

As is well known, there is a correlation between double-difference observations, which are manifested as the correlation between physical observations, and the observations of different epochs are independent of each other. Independent characteristics are one of the necessary conditions for solving using the Kalman filter. Based on the advantages of the Kalman filter in data processing, such as unbiased minimum variance estimation, and considering that the integer ambiguity of the same satellite is equal when the phase observation has no cycle slip, Equation (3) can be reformulated using the Kalman filter, and its filtering equation is:(4)$$\left.\begin{array}{c} X_{k}=\Phi X_{k-1}+W_{k-1}\\ Z_{k}=HX_{k}+V_{k} \end{array}\right\}$$
where $X_{k}$ is the estimated value of the ambiguity at the epoch $t_{k}$, $X_{k}=\nabla ∆N_{k}$. $Z_{k}$ is the observation value at the epoch $t_{k}$, $Z_{k}=Q_{2}^{T}\nabla ∆L$. Φ=1, $H=Q_{2}^{T}B$. $W_{k-1}$, and $V_{k}$ are the white noise sequence, respectively. The float solution of ambiguity and its variance matrix can be solved using the filtering equation, compared with the least squares estimation method of Equation (2), the computational complexity is reduced.

### 2.2. Ambiguity Decorrelation Processing Based on Continuous Cholesky Decomposition

The double-difference processing mode provides a good correlation between the float solution of ambiguity. In the commonly used ambiguity resolution method of LAMBDA, the upper and lower triangular matrix decomposition is alternately selected for the variance matrix of ambiguity, and multiple integer Gaussian transform iterations are performed to obtain the Z-transform matrix for the purpose of decorrelation [10,11,24]. Although the LAMBDA method has a relatively complete mathematical theory basis, the processing and analysis process is more complicated. The ambiguity decorrelation processing based on Cholesky decomposition can improve the computational efficiency [25]. The main calculation process is the integer upper triangular decomposition $UDU^{T}$ and lower triangular decomposition $LDL^{T}$ for the variance matrix of ambiguity. The unit upper triangular matrix U and the unit lower triangular matrix L must satisfy the integer constraint. During the process of iterative decomposition for the variance matrix of ambiguity, the values of the diagonal elements at the front of the diagonal matrix D may be much smaller or much larger than those at the back, resulting in a very large condition number for the matrix D, which is not conducive to subsequent cyclic calculations. In addition, when the iterative decomposition is performed, it is very likely that the value of the diagonal element is close to 0, resulting in the failure of subsequent upper and lower triangular decomposition. Therefore, if the diagonal elements of the matrix are reordered in the repeated iterative decomposition, the problem of the increasing gap between the diagonal elements of the matrix in the decomposition can be avoided, the success rate of the decorrelation decomposition algorithm can be increased, and the efficiency and quality of the ambiguity search can be improved. The idea of the sorting algorithm can be expressed as follows: the descending adjustment matrix is multiplied by the variance matrix of ambiguity, and the diagonal elements of the variance matrix are adjusted in the order from large to small, and then the upper triangular Cholesky decomposition is performed. Subsequently, the ascending adjustment matrix is used to adjust and transform the variance matrix after the upper triangular decomposition, so that the values of the diagonal elements in the transformed new matrix satisfy the rule of increasing from small to large, which can be expressed as follows:(5)$$\overset{⃡}{Q}=SQS^{T}$$
where $\overset{⃡}{Q}$ is the sorted matrix, when the upper triangular Cholesky decomposition of matrix $\overset{⃡}{Q}$ is performed, S is the descending order adjustment matrix, when the matrix $\overset{⃡}{Q}$ performs the lower triangular Cholesky decomposition, S is the ascending adjustment matrix.

Based on the above, the continuous Cholesky decomposition process for ambiguity decorrelation can be expressed as the following three steps:

1. Using the descending order adjustment matrix $S_{1}$ to transform the variance matrix of ambiguity $Q_{\hat{N}}$ to obtain the matrix $Q_{S1}$, the upper triangular Cholesky decomposition is performed on the $Q_{S1}$, and the upper triangular matrix $U_{1}$ is rounded, and then inversed to obtain the integer transformation matrix ${\bar{U}}_{1}^{-1}$, ${\bar{U}}_{1}^{-1}$ can be used to calculate the variance matrix ${\bar{Q}}_{\hat{U}}$ after integer transformation as follows:(6)$${\bar{Q}}_{\hat{U}}={\bar{U}}_{1}^{-1}S_{1}Q_{\hat{N}}S_{1}^{T}{\bar{U}}_{1}^{-T}$$

2. The ascending adjustment matrix $S_{2}$ is used to reorder ${\bar{Q}}_{\hat{U}}$ to obtain the matrix $Q_{S2}$, and the lower triangular Cholesky decomposition is performed on $Q_{S2}$. Subsequently, the lower triangular matrix $L_{1}$ is rounded and then inversed to obtain the transformation matrix ${\bar{L}}_{1}^{-1}$, and the transformed variance matrix ${\bar{Q}}_{\hat{L}}$ can be calculated as follows:(7)$${\bar{Q}}_{\hat{L}}={\bar{L}}_{1}^{-1}S_{2}{\bar{Q}}_{\hat{U}}S_{2}^{T}{\bar{L}}_{1}^{-T}$$

3. Based on the above calculation steps, when the *k*-step iteration is performed and ${\bar{L}}_{k}^{-1}$ is checked as the unit matrix, the final integer invertible ambiguity transformation matrix is:(8)$$T=\left({\bar{L}}_{k}^{-1}S_{2}^{k}{\bar{U}}_{k}^{-1}S_{1}^{k}\right)\left({\bar{L}}_{k-1}^{-1}S_{2}^{k-1}{\bar{U}}_{k-1}^{-1}S_{1}^{k-1}\right)\cdots \left({\bar{L}}_{1}^{-1}S_{2}^{1}{\bar{U}}_{1}^{-1}S_{1}^{1}\right)$$

Accordingly, the variance matrix of ambiguity and float solution after decorrelation can be obtained by transformation matrix T is:(9)$$\left\{\begin{array}{c} Q_{\hat{Z}}=TQ_{\hat{N}}T^{T}\\ \hat{Z}=T\hat{N}\; \end{array}\right.$$

### 2.3. Integer Ambiguity Search Based on Ant Colony Algorithm

After the above integer invertible transformation, the correlation between the variance matrix of ambiguity is reduced, and the ambiguity search problem becomes the following integer least squares problem [13,26].(10)$$fun\left(Z\right)=min{\left\|\hat{Z}-Z\right\|}_{Q_{\hat{Z}}}^{2}=min{\left(\hat{Z}-Z\right)}^{T}Q_{\hat{Z}}\left(\hat{Z}-Z\right)$$

For the integer ambiguity search problem of Equation (9), the ant colony algorithm is introduced into the integer ambiguity search. Based on the principle of positive feedback, the algorithm can quickly find the optimal solution [18,19]. In Equation (10), assuming that the dimension of ambiguity $\hat{Z}$ is *m*, and *P* is the search space of $\hat{Z}$, Equation (10) can be described as finding the parameter vector in the m-dimensional space of *P* that minimizes the value of the objective function $fun\left(Z\right)$, Equation (10) is a continuous variable optimization problem, and ant colony algorithm is a powerful tool for solving combinatorial optimization problems. Therefore, Equation (10) needs to be transformed into a combinatorial optimization problem.

The space of *P* is described as a hyperspace geometry form, the search space contains the optimal integer solution and minimizes the search space. The search space can be defined as follows:(11)$$P=\left\{\left[{\hat{Z}}_{i-}\right]<\left[{\hat{Z}}_{i}\right]<\left[{\hat{Z}}_{i+}\right];i=1,2,\cdots ,b\right\}$$
where $\left[{\hat{Z}}_{i}\right]$ is the i-th integer value of the float solution ${\hat{Z}}_{i}$ after decorrelation, $\left[{\hat{Z}}_{i-}\right]$ and $\left[{\hat{Z}}_{i+}\right]$ are the lower and upper bounds of $\left[{\hat{Z}}_{i}\right]$, respectively, which can be determined by the ratio of the GPS baseline length to the wavelength of satellite signal. The range of $\left[{\hat{Z}}_{i}\right]$ is divided into $k_{i}$ segments, that is, the search space P of the ambiguity parameter to be searched is divided into a discrete space containing a finite number of vectors. The combination of parameters has a total of $A=k_{1}\cdot k_{2}\cdots k_{n}$ cases, so that Equation (10) becomes a combinatorial optimization problem, and finds an optimal ambiguity parameter combination (i.e., shortest path of ant) $Z_{0}$ in A parameter combination (corresponding to  A path of ant), which minimizes the value of the objective function $fun\left(Z\right)$.

Based on the above, the ant colony algorithm for the integer ambiguity search is established, the main calculation steps are as follows:

1. According to Equation (11), the search space of each integer ambiguity parameter and its lower and upper bounds are constructed.

2. At the start, randomly place m ants into the search space, and set the initial pheromone concentration $\tau_{i,j}\left(0\right)=C$ (C is constant). Ants will select an integer from each dimension in their search for each path, and when all dimensions are completed, an array of n integers is formed, representing the end of a loop.

3. Ant k(k=1,2,⋯,m) determines its next movement direction according to the pheromones $\tau_{i,j}\left(t\right)\;$ and visibility $\eta_{i,j}(t)$ on each side during its movement. The probability $P_{i,j}^{k}\left(t\right)$ of ant k transferring from identification parameter i to parameter j is calculated as:(12)$$P_{i,j}^{k}\left(t\right)=\left\{\begin{array}{c} \frac{\tau_{i,j}^{\alpha }\left(t\right)\eta_{i,j}^{\beta }\left(t\right)}{\sum_{k\epsilon {allowed}_{k}}\tau_{i,k}^{\alpha }\left(t\right)\eta_{i,k}^{\beta }\left(t\right)}\;j\epsilon {allowed}_{k}\\ \;0\;\;\;\;\;else \end{array}\right.$$
where ${allowed}_{k}$ is the parameter number that the current ant k can choose to move towards, α is the pheromone concentration factor, and β is the expected heuristic factor. Due to the same visibility among the integer values of ambiguity, the visibility $\eta_{i,j}(t)$ can be expressed as a constant to reduce computational complexity. After the ant colony completes a loop, the pheromone concentration on each path is updated as follows:(13)$$\tau_{i,j}\left(t+∆t\right)=\rho \tau_{i,j}\left(t\right)+\sum_{k=1}^{m}{∆\tau }_{i,j}^{k}$$
where ρ is the pheromone residue coefficient (0≤ρ≤1), ${∆\tau }_{i,j}^{k}$ is the pheromone concentration left by ant k between parameters i and j, and ${∆\tau }_{i,j}^{k}$ is expressed as:(14)$${∆\tau }_{i,j}^{k}=\left\{\begin{array}{c} \frac{Q}{fun(Z_{k})}\;\left(Ant\;k\;chooses\;path\;i,\;j\right)\\ \;0\;\;\;\;\;\left(else\right)\; \end{array}\right.$$
where Q is the constant of pheromone strength, $Z_{k}$ is the integer value array of ambiguities searched by ant k, and $fun\left(Z_{k}\right)\;$ is the objective function value corresponding to the integer value array of ambiguities selected by ant k in this loop. The minimum value of this objective function represents that the searched integer array is the optimal solution.

4. Determine whether the number of iterations meets the preset maximum number of iterations $N_{max}$, if it does, stop the search and output the optimal integer array. Otherwise, continue with steps (2) to (4). In the subsequent experiments of this paper, the main parameters of the ant colony algorithm are shown in Table 1.

## 3. Results

In this section, the GPS static and dynamic measured data are used as examples to validate the above ambiguity resolution method. The feasibility and correctness of this method are analyzed based on the deviation between the float solution of ambiguity and its standard value, and the success rate of ambiguity resolution and the information bias of positioning.

### 3.1. GPS Static Data Experiment

The GPS dual-frequency measured data with a baseline length of about 254.4 m are used, and its sampling rate is 1 s, eight GPS satellites were continuously tracked, with a total of 5001 epochs of data were collected. The RTKLIB software is used to solve these data epoch by epoch to obtain the three-dimensional coordinates of the non-reference station [27]. Based on this, the integer ambiguity of seven satellites for G26-G29, G26-G31, G26-G14, G26-G32, G26-G22, G26-G03, and G26-G16 is determined, that is, the integer ambiguity of L1 frequency $N_{L1}=$ [−10, −15, −8, −1, −11, −8, −10]^T^, and the integer ambiguity of L2 frequency $N_{L2}=$ [1, −7, −5, −2, −4, −5, −7]^T^ as standard values. In this experiment, two different data solution schemes are designed to solve.

1. The first scheme. Based on the pseudorange and phase observation data, which is solved by the conventional least squares and LAMBDA method.

2. The second scheme. Using only the phase observation data, the new method of integer ambiguity resolution based on eliminating coordinate parameters and ant colony algorithm is solved by epoch.

The deviation between the float solution and the standard value of integer ambiguity $N_{L1}\;$ and $N_{L2}$ obtained by two different experimental schemes are calculated, respectively, and the statistical histograms of the differences are shown in Figure 1 and Figure 2. Meanwhile, some of the $N_{L1}$ and $N_{L2}$ ambiguity parameters in the consecutive epoch calculated by the two schemes are selected and listed in Table 2.

From Figure 1 and Table 2, it can be seen that the accuracy of the float solution of ambiguity using the least squares method solved epoch by epoch in the first scheme is very poor. The integer ambiguity for each satellite obtained by the LAMBDA method deviates greatly from its standard value, and the success rate of the correctly fixed integer ambiguity is 0%. At the same time, from Figure 2 and Table 2, the difference between the float solution of ambiguity and its standard value calculated by the new algorithm of GPS ambiguity resolution proposed in this paper is mostly within ±0.5 cycles, which is very close to the standard value of ambiguity. It shows that the method can obtain the high-precision float solution of ambiguity by using the Kalman filter after the elimination of coordinate parameters through the *QR* decomposition. The new method searches for the fixed solution of integer ambiguity of each satellite in the same way as the standard value, which significantly improves the accuracy of fixing integer ambiguity.

In addition, the integer ambiguity fixed by the new method in the second scheme is used to calculate the three-dimensional coordinates of the non-reference station epoch by epoch. The results are compared with the three-dimensional coordinates calculated by RTKLIB. The residual sequences of coordinate in the X, Y, and Z directions for each epoch are shown in Figure 3.

From the residual sequence of coordinate in Figure 3, it can be seen that the overall residual values of all epochs are small, indicating that the ambiguity is correctly fixed. Meanwhile, the positioning accuracy of the method in solving coordinates for GPS static data are statistically analyzed. The RMSE in X, Y, and Z direction is $\sigma_{X}$ = ±3.2 mm, $\sigma_{Y}$ = ±9.5 mm, and $\sigma_{Z}$ = ±4.6 mm, respectively, which verified the feasibility of the new method.

### 3.2. GPS Dynamic Data Experiment

During the experimental process of obtaining this dynamic data, the sampling rate of both the reference station and the rover station was 0.2 s, seven GPS satellites were continuously tracked, and a total of 9122 epochs of dynamic data were collected. The GPS dynamic data are resolved using RTKLIB software, the three-dimensional coordinates (X, Y, Z) for each epoch of the rover station are obtained, and its motion trajectory is shown in Figure 4. The integer ambiguity $N_{L1}$ = [1, 28, 8, −3, 6, −7]^T^, $N_{L2}$ = [4, 43, 5, −1, 3, −14]^T^ of six satellites obtained by RTKLIB for G31-G29, G31-G16, G31-G30, G31-G32, G31-G20, and G31-G14 are used as the standard values. In order to further verify the accuracy of the float solution of ambiguity calculated by the ambiguity resolution method proposed in this paper, the ambiguity results obtained by the new method are compared with the standard value of ambiguity by RTKLIB. The statistical histogram of the difference between the float solution of ambiguity and the standard values for each satellite are shown in Figure 5.

In addition, based on the least squares and LAMBDA method of the first scheme mentioned above to solve the float solution of ambiguity of each epoch for each satellite, the results are subtracted from the standard values of ambiguity to obtain a statistical histogram as shown in Figure 6. The information parameters of $N_{L1}$ and $N_{L2}$ ambiguity for partial continuous epoch solved by the first scheme and the new method (the second scheme) are listed in Table 3.

As can be seen from Figure 5 and Table 3, for GPS dynamic data, the number of epochs with the difference between the float solution of ambiguity solved by the new method and the standard value near 0 cycles is the largest, and the overall trend is normal distribution, the result of the float solution of ambiguity is close to the standard value of ambiguity, which is beneficial to the accurate calculation of integer ambiguity. However, in Figure 6 and Table 3, it is also shown that the float solution of ambiguity solved by the first scheme deviates seriously from its standard value, and its difference fluctuates greatly. The ambiguity cannot be fixed correctly even if the ambiguity resolution method LAMBDA is used, and the success rate of ambiguity resolution is 0%.

At the same time, the new method is used to solve the coordinates in the X, Y, and Z direction of each epoch for the rover station, and the corresponding coordinate residual sequence is obtained by comparing with the three-dimensional coordinates of the rover station obtained by RTKLIB, as shown in Figure 7. It can be found that the overall coordinate residual values of all epochs are small. The positioning accuracy of the coordinates solved by the new method for GPS dynamic data are $\sigma_{X}$ = ±3.6 mm, $\sigma_{Y}$ = ±4.8 mm, and $\sigma_{Z}$ = ±2.5 mm, which further verifies the effectiveness of the method.

## 4. Discussion

After obtaining the float solution of ambiguity and its variance matrix by eliminating the coordinate parameters and solving with the Kalman filter, in order to evaluate the effectiveness of the algorithm in ambiguity decorrelation processing, the condition numbers κ and the decorrelation coefficient γ are used as quantitative evaluation indexes, and the corresponding calculation expression are as follows [12,22]:(15)$$\kappa =\left|\lambda_{max}\left(Q_{\hat{Z}}\right)/\lambda_{min}\left(Q_{\hat{Z}}\right)\right|$$



(16)
 γ=detRZ^12RZ^=diagQZ^−12·QZ^·diagQZ^−12



In Equations (15) and (16), where $Q_{\hat{Z}}$ is the variance matrix of ambiguity, $\lambda_{max}$ and $\lambda_{min}$ denote the maximum and minimum eigenvalues of $Q_{\hat{Z}}$, respectively, $det\left[\cdot \right]$ is a matrix determinant operation.

Under the two sets of GPS static data and dynamic data, the condition numbers and decorrelation coefficient information of the original variance matrix and the transformed matrix after continuous Cholesky decomposition decorrelation processing are shown in Figure 8 and Figure 9, respectively. From Figure 8 and Figure 9, it can be seen that the condition number of the original variance matrix of ambiguity before decorrelation is large, and the decorrelation coefficient is very small (close to 0). Due to the geometric meaning of the condition number representing the flattening of the hyper-ellipsoid, the decorrelation coefficient $\gamma \left(\gamma \in \left[0,1\right]\right)$ mathematically characterizes the decorrelation effect, and the larger the value of γ, the lower the correlation of the matrix. Therefore, the flattening of the search ellipsoid based on the original variance matrix of the ambiguities is very large, indicating that there is a strong correlation between ambiguities. After the decorrelation processing of the ambiguity variance matrix, the flattening rate of the corresponding search ellipsoid is greatly reduced, and the value of decorrelation coefficient is evidently increased. This indicates that the method has good applicability in the decorrelation processing of the variance matrix, and the good decorrelation effect can be obtained.

In addition, in order to further verify the effectiveness of using the ant colony algorithm to search the integer ambiguity, Equation (17) is used to calculate the fixed success rate of integer ambiguity for GPS static and dynamic data [28], and the statistical information is shown in Table 4.(17)$$P_{FN}=\frac{N_{FN}}{N_{T}}\times 100\%,$$
where $P_{FN}$ is the fixed success rate of integer ambiguity, $N_{FN}$ is the number of epochs in which the ambiguity of all satellites is fixed to the correct value, $N_{T}$ is the total number of epochs for ambiguity resolution.

It can be seen from Table 4 that the fixed success rate of integer ambiguity using the ant colony algorithm for two different experiments has reached over 99%, which indicates that the ant colony algorithm has high feasibility in integer ambiguity search.

## 5. Conclusions

In this paper, two sets of GPS static and dynamic measured data are analyzed. The experimental results show that coordinate parameter information can be eliminated using *QR* decomposition and the corresponding transformation on the coefficient matrix of coordinate in the double-difference observation equation. The establishment of the Kalman filter recursive equation, with only an ambiguity parameter for solving, can not only reduce the computational burden of unknown parameters, but also improve the accuracy of float solution of ambiguity. At the same time, in the process of continuous Cholesky decomposition of variance matrix of ambiguity, the variance matrix is decomposed and transformed after descending and ascending, respectively, which can realize the maximum uncorrelation in the ambiguity search space and achieve a good decorrelation effect. On this basis, the ant colony algorithm is used to search the integer ambiguity, which can obtain a higher success rate. It is verified that the new method for solving GPS integer ambiguity proposed in this paper is correct and effective.

## Figures and Tables

**Figure 1 sensors-25-00321-f001:**
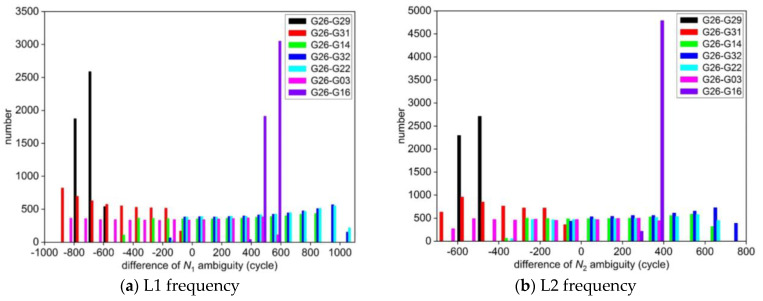
Statistics of the deviation between the float solution and the standard value of ambiguity for each satellite in the first scheme.

**Figure 2 sensors-25-00321-f002:**
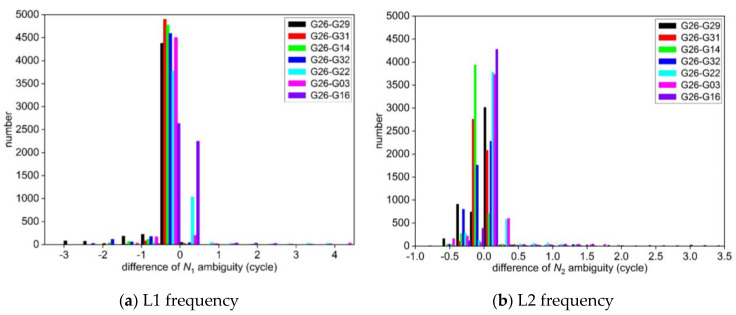
Statistics of the deviation between the float solution and the standard value of ambiguity for each satellite in the second scheme.

**Figure 3 sensors-25-00321-f003:**
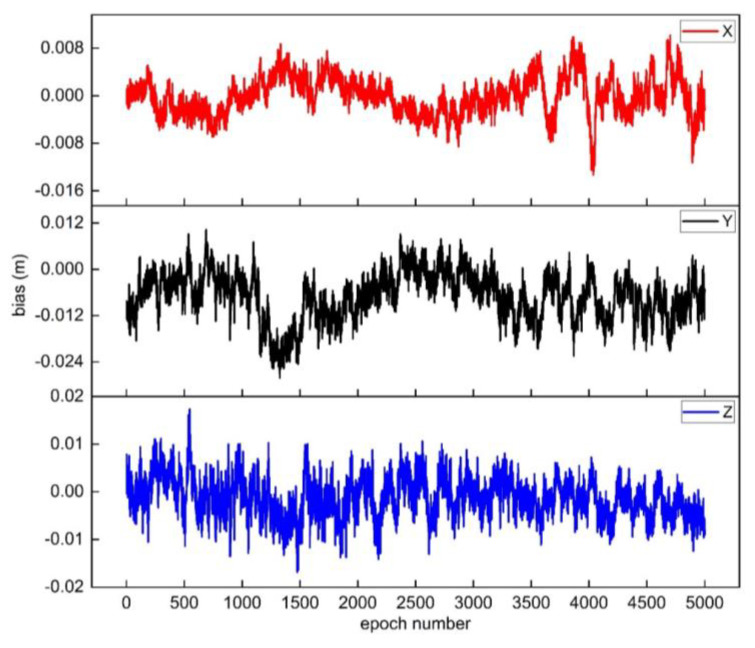
The residual sequences of coordinate for each epoch in the GPS static data experiment.

**Figure 4 sensors-25-00321-f004:**
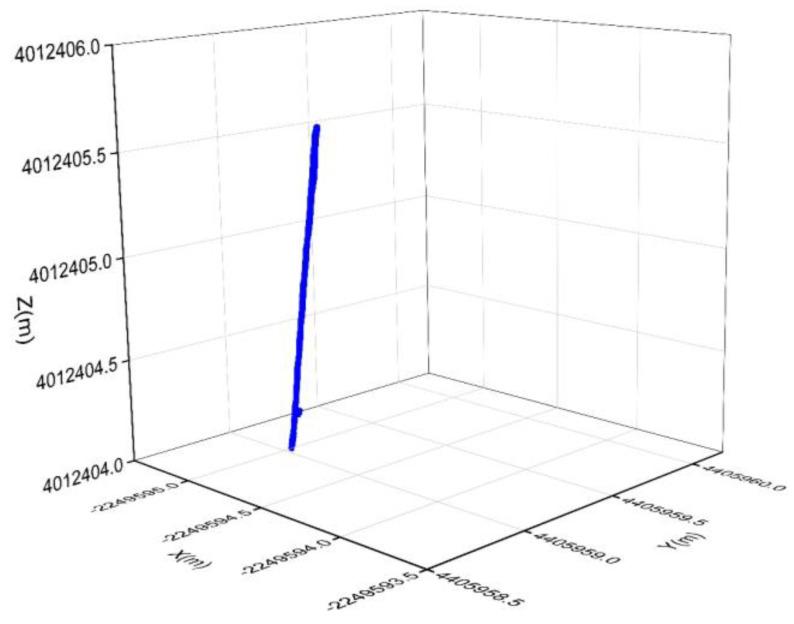
The motion trajectory of rover station.

**Figure 5 sensors-25-00321-f005:**
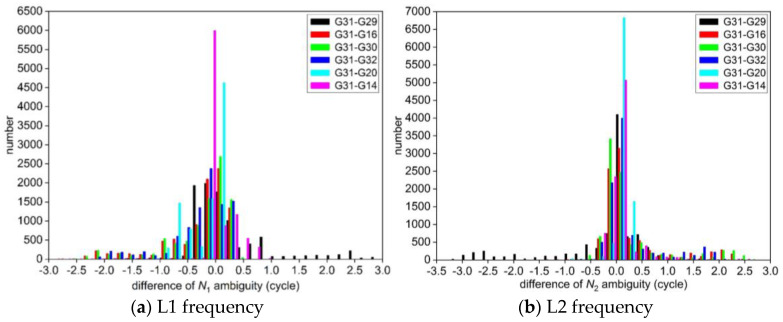
Statistics of the deviation between the float solution and the standard value of ambiguity for each satellite by the new method.

**Figure 6 sensors-25-00321-f006:**
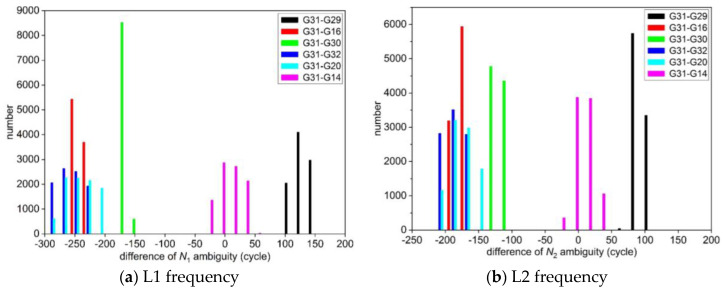
Statistics of the deviation between the float solution and the standard value of ambiguity for each satellite in the first scheme.

**Figure 7 sensors-25-00321-f007:**
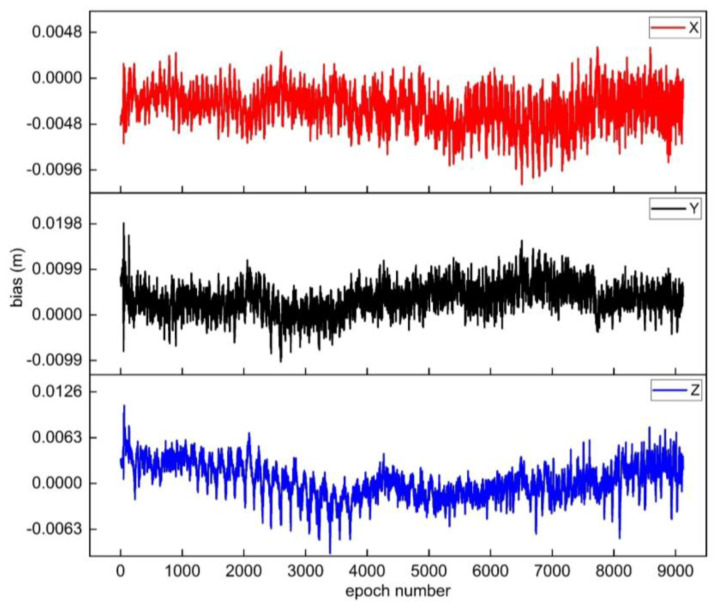
The residual sequences of coordinate for each epoch in the GPS dynamic data.

**Figure 8 sensors-25-00321-f008:**
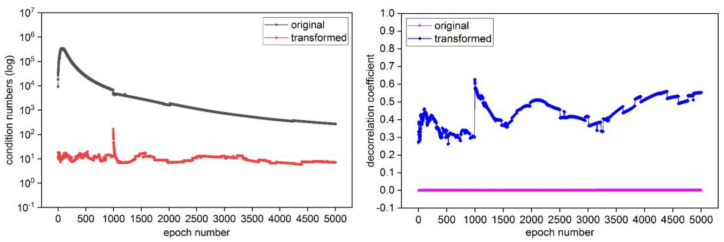
The condition numbers and decorrelation coefficient in the GPS static data experiment.

**Figure 9 sensors-25-00321-f009:**
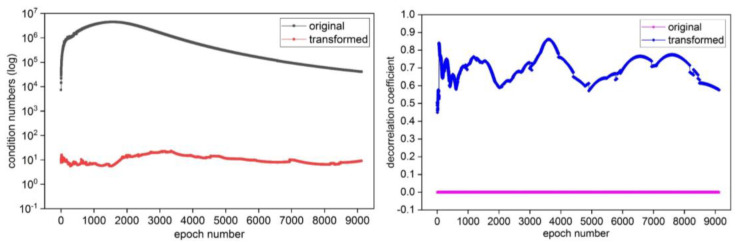
The condition numbers and decorrelation coefficient in the GPS dynamic data experiment.

**Table 1 sensors-25-00321-t001:** The main parameter setting of ant colony algorithm.

Number of Ants (*m*)	Pheromone Concentration Factor (*α*)	Expected Heuristic Factor (*β*)	Pheromone Residue Coefficient (*ρ*)	Constant of Pheromone Strength (*Q*)	Maximum Number of Iterations (*N_max_*)
20	1	2	0.8	100	50

**Table 2 sensors-25-00321-t002:** Comparison of $N_{L1}$ and $N_{L2}$ ambiguity parameters solved by two schemes.

Number	The First Scheme		The Second Scheme	
	Integer Solution of $\mathrm{Ambiguity}\;{(N}_{L1})$	Integer Solution of $\mathrm{Ambiguity}\;{(N}_{L2})$	Integer Solution of $\mathrm{Ambiguity}\;{(N}_{L1})$	Integer Solution of $\mathrm{Ambiguity}\;{(N}_{L2})$
1	−733, −790, −161, 143, −146, −625, 448	−529, −568, −117, 97, −105, −455, 320	−10, −15, −8, −1, −11, −8, −10	1, −7, −5, −2, −4, −5, −7
2	−736, −789, −160, 141, −151, −630, 443	−533, −571, −119, 96, −102, −453, 323	−10, −15, −8, −1, −11, −8, −10	1, −7, −5, −2, −4, −5, −7
3	−734, −790, −159, 144, −151, −631, 445	−526, −570, −117, 101, −100, −451, 326	−10, −15, −8, −1, −11, −8, −10	1, −7, −5, −2, −4, −5, −7
4	−732, −788, −160, 143, −145, −623, 447	−527, −574, −116, 105, −102, −456, 329	−10, −15, −8, −1, −11, −8, −10	1, −7, −5, −2, −4, −5, −7
5	−736, −793, −159, 146, −150, −632, 448	−528, −574, −118, 103, −97, −450, 331	−10, −15, −8, −1, −11, −8, −10	1, −7, −5, −2, −4, −5, −7
6	−734, −788, −160, 142, −145, −623, 446	−524, −570, −114, 105, −104, −456, 325	−10, −15, −8, −1, −11, −8, −10	1, −7, −5, −2, −4, −5, −7
7	−737, −790, −159, 143, −149, −629, 445	−529, −574, −116, 104, −102, −456, 328	−10, −15, −8, −1, −11, −8, −10	1, −7, −5, −2, −4, −5, −7
8	−741, −791, −161, 140, −146,−626, 446	−525, −569, −115, 103, −101, −452, 325	−10, −15, −8, −1, −11, −8, −10	1, −7, −5, −2, −4, −5, −7
9	−734, −792, −159, 147, −144,−625, 451	−523, −570, −114, 106, −101, −453, 327	−10, −15, −8, −1, −11, −8, −10	1, −7, −5, −2, −4, −5, −7
10	−728, −785, −157, 146, −144, −621, 446	−530, −571, −116, 101, −101, −453, 325	−10, −15, −8, −1, −11, −8, −10	1, −7, −5, −2, −4, −5, −7

**Table 3 sensors-25-00321-t003:** Comparison of $N_{L1}\;$ and $N_{L2}$ ambiguity parameters solved by two schemes.

Number	The First Scheme		The Second Scheme	
	Integer Solution of $\mathrm{Ambiguity}\;{(N}_{L1})$	Integer Solution of $\mathrm{Ambiguity}\;{(N}_{L2})$	Integer Solution of $\mathrm{Ambiguity}\;{(N}_{L1})$	Integer Solution of $\mathrm{Ambiguity}\;{(N}_{L2})$
1	153, −202, −155, −235, −204, 21	112, −124, −112, −170, −152, 6	1, 28, 8, −3, 6, −7	4, 43, 5, −1, 3, −14
2	150, −200, −153, −233, −203, 20	109, −131, −116, −173, −153, −1	1, 28, 8, −3, 6, −7	4, 43, 5, −1, 3, −14
3	152, −205, −156, −238, −208, 20	113, −123, −111, −171, −155, 9	1, 28, 8, −3, 6, −7	4, 43, 5, −1, 3, −14
4	152, −205, −156, −238, −208, 20	111, −124, −111, −171, −155, 7	1, 28, 8, −3, 6, −7	4, 43, 5, −1, 3, −14
5	152, −205, −156, −238, −208, 20	118, −117, −110, −166, −147, 13	1, 28, 8, −3, 6, −7	4, 43, 5, −1, 3, −14
6	151, −205, −155, −239, −211, 21	116, −118, −110, −166, −147, 11	1, 28, 8, −3, 6, −7	4, 43, 5, −1, 3, −14
7	151, −205, −155, −239, −211, 21	110, −130, −115, −174, −156, 2	1, 28, 8, −3, 6, −7	4, 43, 5, −1, 3, −14
8	152, −205, −156, −238, −208, 20	114, −127, −115, −172, −152, 5	1, 28, 8, −3, 6, −7	4, 43, 5, −1, 3, −14
9	152, −205, −156, −238, −208, 20	118, −117, −110, −166, −147, 13	1, 28, 8, −3, 6, −7	4, 43, 5, −1, 3, −14
10	156, −195, −151, −232, −203, 28	112, −124, −112, −170, −152, 6	1, 28, 8, −3, 6, −7	4, 43, 5, −1, 3, −14

**Table 4 sensors-25-00321-t004:** The success rate of ant colony algorithm for searching integer ambiguity.

Experimental Data	Number of Epochs for Correct Integer Ambiguity Searching	Total Number of Epochs	The Success Rate for Searching
GPS static data	5001	5001	100%
GPS dynamic data	9108	9122	99.8%

## Data Availability

The datasets analyzed in this study are managed by the College of Geology Engineering and Geomatics, Chang’an University, and can be available on request from the corresponding author.

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
