# Peer review of "GPS Phase Integer Ambiguity Resolution Based on Eliminating Coordinate Parameters and Ant Colony Algorithm"

_sensors, 2025, doi:10.3390/s25020321_

Round 1

Reviewer 1 Report

Comments and Suggestions for Authors

Dear Authors, your manuscript presents a novel approach to GPS integer ambiguity resolution using QR decomposition and ant colony optimization, with promising results demonstrated through static and dynamic data. While the methodology is detailed and innovative, there are areas where clarity and justification could enhance the impact, particularly regarding assumptions and comparisons to existing methods. Overall, it shows significant potential for contributing to high-precision GPS positioning research. I have marked my comments below, if addressed well, will improve the quality of the manuscript.

Lines 28–68: The literature review is detailed but lacks focus on the novelty of the proposed method. It repeats known details about ant colony optimization (ACO) and ambiguity resolution without specifying why the ACO-based method improves current practices.

The transition from equation (1) to (3) assumes QR decomposition eliminates coordinate parameters but skips explaining the implications for real-world datasets.

Kalman filter assumptions (independence and Gaussian noise) are not justified for GPS double-difference observations. Kindly add justification for assumptions and discuss limitations (e.g., when noise is not Gaussian).

The Cholesky decomposition discussion lacks clarity on computational efficiency improvements over LAMBDA. Provide quantitative comparisons or theoretical bounds demonstrating Cholesky’s advantage in reducing computation time.

The ACO initialization parameters are arbitrarily chosen without a sensitivity analysis. I recommend the authors to include a brief analysis or justification for parameters like pheromone concentration factor and heuristic factor.

Lines 245–276: The histograms lack statistical interpretation. Why does ±0.5 cycles represent an improvement, and what does it imply about the float solution’s reliability?

Lines 296–330: The success rate (99.8%) of the ACO method is impressive, but the paper does not address edge cases where ACO failed.

The discussion is largely descriptive and lacks critical evaluation of the method’s performance compared to alternatives. It ignores scalability for larger datasets or real-time GPS applications. Kindly discuss computational overhead and feasibility for real-time applications.

Author Response

Thank you very much for your letter and the comments about our paper submitted to sensors-3374866. We have learned much from the reviewers’ comments, which are fair, encouraging and constructive. After carefully studying the comments and your advice, we have made corresponding changes. Our response of the comments is enclosed at the end of this letter.Please see the attachment. If you have any question about this paper, please don’t hesitate to contact us.

Reviewer 2 Report

Comments and Suggestions for Authors

The correct fixing of GNSS integer ambiguity is the key to realize the application of GNSS high-precision positioning. In this manuscript, the QR decomposition transformation is carried out on the coordinate parameter coefficient matrix in the double difference observation equation to eliminate the coordinate parameter information, so as to reduce the dimension of the unknown parameters. On the basis of using kalman filter to estimate the floating solution of ambiguity, the continuous Cholesky decomposition is used to reduce the correlation. Then, the ambiguity search problem is transformed into a combinatorial optimization problem, and the artificial intelligence ant colony algorithm is used to search for the optimal integer solution. The static and dynamic measured data are selected, and two different solution schemes are used to verify the proposed integer ambiguity resolution method, the results show that the method is feasible and effective. The method described in this manuscript is detailed and the experimental results are rich, the following problems need to be explained in the manuscript

1. When using kalman filter to estimate only the ambiguity parameters, how to take the initial value of the parameter to be estimated ?

2. In the manuscript, the conventional least squares method in the first scheme is used to solve the N1 and N2 ambiguity parameters by epoch, why is the accuracy for the floating solution of ambiguity very poor ?

3. The GPS observation data used in the experiment should be all short baselines, can the integer ambiguity algorithm proposed in the manuscript be applied to medium-long baselines ?

Author Response

(The authors gave the same response as above.)

Round 2

Reviewer 1 Report

Comments and Suggestions for Authors

The authors have addressed my comments well. I do not have any comments. I recommend its publication.